# Evaluation of Various Starchy Foods: A Systematic Review and Meta-Analysis on Chemical Properties Affecting the Glycemic Index Values Based on In Vitro and In Vivo Experiments

**DOI:** 10.3390/foods10020364

**Published:** 2021-02-08

**Authors:** Frendy Ahmad Afandi, Christofora Hanny Wijaya, Didah Nur Faridah, Nugraha Edhi Suyatma, Anuraga Jayanegara

**Affiliations:** 1Department of Food Science and Technology, IPB University, Bogor 16880, Indonesia; frendy_afandi@apps.ipb.ac.id (F.A.A.); channywijaya@apps.ipb.ac.id (C.H.W.); nugrahaedhis@gmail.com (N.E.S.); 2Deputy Ministry for Food and Agribusiness, Coordinating Ministry for Economic Affairs Republic of Indonesia, Jakarta 10710, Indonesia; 3Department of Animal Nutrition and Feed Technology, IPB University, Bogor 16880, Indonesia; anuraga.jayanegara@gmail.com

**Keywords:** bioactive compounds, carbohydrate foods, diabetes, glycemic index, meta-analysis

## Abstract

The chemical properties that serve as major determinants for the glycemic index (GI) of starchy food and recommended low-GI, carbohydrate-based foods have remained enigmatic. This present work performed a systematic assessment of linkages between chemical properties of foods and GI, and selected low-GI starchy foods. The data were sourced from literature published in various scientific journals. In total, 57 relevant studies and 936 data points were integrated into a database. Both in vitro and in vivo studies on GI values were included. The database was subsequently subjected to a meta-analysis. Meta-analysis from in vitro studies revealed that the two significant factors responsible for the GI of starchy foods were resistant starch and phenolic content (respectively, standardized mean difference (SMD): −2.52, 95% confidence interval (95%CI): −3.29 to −1.75, *p* (*p*-value) < 0.001; SMD: −0.72, 95%CI: −1.26 to −0.17, *p* = 0.005), while the lowest-GI crop type was legumes. Subgroup analysis restricted to the crop species with significant low GI found two crops, i.e., sorghum (SMD: −0.69, 95%CI: −2.33 to 0.96, *p* < 0.001) and red kidney bean (SMD: −0.39, 95%CI: −2.37 to 1.59, *p* = 0.001). Meta-analysis from in vivo studies revealed that the two significant factors responsible for the GI of starchy foods were flavonoid and phenolic content (respectively, SMD: −0.67, 95%CI: −0.87 to −0.47, *p* < 0.001; SMD: −0.63, 95%CI: −1.15 to −0.11, *p* = 0.009), while the lowest-GI crop type was fruit (banana). In conclusion, resistant starch and phenolic content may have a desirable impact on the GI of starchy food, while sorghum and red kidney bean are found to have low GI.

## 1. Introduction

Type 1 diabetes mellitus (T1DM) has become a chronic metabolic disorder world-wide, and the regulation of blood glucose at a near-normal level could best fit the goals of preventing or delaying long-term diabetes complications in T1DM [1]. Insulin treatment alone is inadequate for controlling T1DM; essentially, dietary adjustments are required for the proper regulation of blood glucose level. In addition to type 1 diabetes mellitus, the glycemic index is associated with other non-communicable diseases such as cardiovascular diseases (CVDs), type 2 diabetes, and cancer [2]. Glycemic index (GI) is defined as the blood glucose response measured as the area under the curve (AUC) in response to a test food that an individual consumes under standard conditions and is expressed as a percentage of the AUC following consumption of a reference food that the same individual consumes on a different day [1]. A GI classification system is in common use. In this system, foods are categorized as having low (≤55), medium (55 < GI ≤ 70), or high GI (>70) [2]. Carbohydrates are among the major determinants of postprandial blood glucose and are directly related to the GI. Accordingly, there is a crucial need for information regarding the relationship between various starchy foods and their GI properties. Such information would allow consumers to appropriately adjust their foods based on the GI profile [3,4]. Further, elucidating the chemical properties of starchy foods responsible for GI is essential. These chemical properties can be relevant indicators for selecting foods with low GI [5]. However, the relationship between the chemical properties of starchy foods and GI is not thoroughly known.

Evidence from randomized controlled trials (RCTs) was reported to be inconsistent [4,6,7], while no report was found using systematic review [8] or meta-analysis [9]. Furthermore, five recent trials [10,11,12,13,14] with adequate power have been published and involved new evidence. Therefore, we performed this meta-analysis to explain the chemical property factors affecting the GI of starchy foods and selected low-GI carbohydrate foods.

## 2. Materials and Methods

### 2.1. Literature Search

This research referred to the guidelines of a meta-analysis handbook [15]. Relevant studies published in various scientific journals and indexed in electronic databases such as PubMed, Proquest, Science Direct, Cengage Library, and Google Scholar were identified, focusing on the relationship between food chemical properties and GI values.

Keywords used in the search strategy included “chemical properties”, “glycemic index”, “carbohydrate category”, and/or “blood glucose response”. After examining the titles and abstracts, we excluded irrelevant studies. Subsequently, we examined the full texts of all remaining articles to determine eligibility. The discrepancies were verified by discussion and consensus. We also reviewed the identified trials and review articles in reference lists to find any other potential proper reports.

This systematic review and meta-analysis were based on the PRISMA (Preferred Reporting Items for Systematic Reviews and Meta-Analyses statement) checklist. Each section and each subsection should be clearly identified. The search strategy used in the electronic databases employed keywords, inclusion criteria, and exclusion criteria. The keywords that were used in this research were: available carbohydrate, glycemic index, blood glucose, diabetic, weight management, and hypoglycemic. Among those keywords, only glycemic index was detected in the MeSH (Medical Subject Headings) database. The inclusion criteria were the following: peer-reviewed, clinical trial, completely randomized design, in vitro, in vivo, within the last 20 years. Exclusion criteria were the following: letters to the editor, proceedings, abstracts, and book chapters.

### 2.2. Eligibility Criteria

The eligibility of the trials was determined according to the following criteria: (1) design: completely randomized design that analyzed the relationships between food chemical properties and GI values; (2) population: all research that applied both in vitro and in vivo protocols for determining GI; (3) intervention: comparison between chemical properties, GI value, and selected low-GI carbohydrate-based foods; and (4) data: sufficient information (data) for calculating the standardized mean difference (SMD) and the corresponding 95% confidence interval (CI). Further, all published papers were written in English.

### 2.3. Data Extraction

Data from each included study were extracted and integrated into the database. The following data were collected: authors, year of publication, country of origin, number of patients or participants, intervention, control, sample/type, method/reference food, outcomes data (GI value), and follow-up.

### 2.4. Risk of Bias Assessment

To assess the risk of bias, Cochrane risk of bias tool was employed [16]. This was performed by using the following criteria: allocation concealment; random sequence generation; participants and personnel blinding; outcome assessment blinding; selective reporting; incomplete outcome data; and other bias. Each study was evaluated and scored as having “high”, “low”, or “unclear” risk of bias. Patient and clinician blinding was highly difficult and generally not feasible in these tests, and we determined that the main outcome was less vulnerable to being affected by lack of blinding. Consequently, studies with a high risk of bias for any one (or more) of the key domains were considered to have a high risk of bias. Studies for all key domains except blinding were considered to have a low risk of bias; otherwise, studies were considered to have an unclear risk of bias [17].

### 2.5. Quality of Evidence Assessment

The quality of evidence for primary and secondary outcomes was evaluated according to GRADE (Grading of Recommendations, Assessment, Development, and Evaluation) procedure for risk of bias, inconsistency, indirectness, imprecision, and publication bias, which were categorized as high, moderate, low, or very low [18]. The results were then summarized in tables constructed using the GRADE system [18,19,20] (GRADE version 3.6) (Table 1). Stages of the study, including literature search, data extraction, risk of bias assessment, and evidence grade assessment were performed independently by one author (Frendy Ahmad Afandi/FAA).

### 2.6. Statistical Analysis

We used weighted analysis using Hedges’ d (Standard Mean Difference/SMD) for statistical methods. The data extracted from selected journals were mean, standard deviation or standard error, and the number of replicate experiments. The SMD with corresponding 95%CI values were pooled using the random-effects model. Exploration of heterogeneity across studies was carried out using the I^2^ index [20] (the I^2^ > 50% indicated sufficient heterogeneity), and publication bias was determined using the Begg’s test and Egger’s test (*p* < 0.05 was considered statistically significant for publication bias). We used Meta-Essentials tools for the meta-analysis process. The criterion of publication bias assessment approach using the GRADE System, consisted of selection, performance, attrition, detection, and reporting bias. The variables used for subgroup analysis were the method of study (in vivo or in vitro), type of reference food, and type of crops.

## 3. Results

### 3.1. Trial Selection and Risk of Bias Assessment

A total of sixty articles (from 338 articles) were selected for full-text review, resulting in fifty-seven articles [17,18,19,20,21,22,23,24,25,26,27,28,29,30,31,32,33,34,35,36,37,38,39,40,41,42,43,44,45,46,47,48,49,50,51,52,53,54,55,56,57,58,59,60,61] that met the inclusion criteria. Eight of them were rejected due to a lack of essential data. Five additional articles [25,26,47,50,51] from reference lists of identified trials were included in the study because they met the inclusion criteria. In total, the meta-analysis involved fifty-seven articles, as exhibited in Figure 1. According to the Cochrane Collaboration tool, eleven trials [6,24,29,32,33,37,38,42,48,51,54] were categorized as being at a low risk of bias, while forty-four were categorized as unclear [17,19,20,21,22,23,25,27,28,30,31,34,35,36,39,40,41,43,44,45,46,47,49,50,52,53,55,56,57,58,59,60,61,62], and two were categorized as being at a high risk of bias [18,26]. Details about the risk of bias are supplied in Figure 2 and Figure 3.

### 3.2. Characteristics of Articles

Fifty-seven studies, involving 936 participants, were published from 2002 to 2019. Twenty-six studies [9,10,11,12,17,18,20,21,26,31,35,39,43,44,45,46,47,49,54,55,56,57,59,60,61,62] used an in vitro experiment, while thirty-two studies involved in vivo experiments. Only one study used both in vitro and in vivo data experiments [11]. The in vivo studies involved healthy participants, and some studies used rats [51,58]. The PICOS of this research is defined as Participants, Interventions, Comparisons, Outcomes, and Study Design. Participants in the in vivo experiment were healthy adults. Interventions were lower food chemical properties. Comparisons were higher food chemical properties. Outcomes of this research were resistant starch, dietary fibre, protein, phenolic, and flavonoid content significantly affecting GI starchy foods, while selected low-GI foods were sorghum and red kidney bean. The study design used in this research was the completely randomized design. Among fifty-seven included studies, forty-eight were selected for discussion of the relationship between chemical properties and GI value [17,18,19,20,21,22,23,24,25,26,27,28,29,30,31,32,33,34,35,36,37,38,39,40,41,42,43,44,45,46,47,48,49,50,51,52,53,54]. The remaining studies were used for the selection of low-GI carbohydrate-based foods [55,56,57,58,59,60,61,62]. All studies reported changes in glycemic index, four studies [7,17,18,19] reported changes in amylose content to GI, ten studies [9,10,11,12,13,20,21,22,23,24] reported changes in resistant starch (RS) content to GI, fifteen studies reported a change in dietary fibre content to GI, fifteen studies reported a change in fat content to GI, fifteen studies reported a change in protein content to GI, twelve studies reported a change in phenol content to GI, ten studies reported a change in flavonoid content to GI, five studies reported a change in cereal type to GI, six studies reported a change in tuber type to GI, two studies reported a change in fruit type to GI, and four studies reported a change in legume type to GI. Detailed characteristics of eligible studies are presented in Table 2 and Table 3.

### 3.3. Primary Outcome

The primary outcome was the chemical properties determinant responsible for the GI of starchy foods and selected low-GI carbohydrate-based foods in the forest plots and funnel plots graphic (Figure 4, Figure 5a, Figure 6, and Figure 7a). All studies (a total of 936 participants) provided data on the determinant of chemical properties (912 participants) and selected low-GI carbohydrate-based foods (twenty-four participants). Compared to other chemical properties, resistant starch and phenolic content significantly reduced the GI value of starchy foods (respectively, SMD: −2.52, 95%CI: −3.29 to −1.75, *p* < 0.001; SMD: −0.72, 95%CI: −1.26 to −0.17, *p* = 0.005) with heterogeneity (respectively, I^2^ = 84.23%, *p* < 0.001; I^2^ = 73.64%, *p* = 0.005) (Table 2, Table 4, and Table 5). The lowest GI crop type is legumes (SMD: −2.15, 95%CI: −3.45 to −0.85, *p* < 0.001) with heterogeneity (I^2^ = 72.97%, *p* < 0.001) (Figure 7). The heterogeneity among these studies may result from discrepancies in the population and control group.

### 3.4. Subgroup Analysis

Crop species, i.e., sorghum and red kidney bean, had low GI (respectively, SMD: −0.69, 95%CI: −2.33 to 0.96, *p* < 0.001; SMD: −0.39, 95%CI: −2.37 to 1.59, *p* = 0.001), with heterogeneity (respectively, I^2^ = 81.4%, *p* < 0.001; I^2^ = 73.0%, *p* = 0.001). All results of subgroup analyses are presented in Table 5 and Figure 7.

### 3.5. Secondary Outcomes

The contribution of six chemical properties to GI and five source types of carbohydrates to GI can be seen in Table 4 and Table 5. Forty-eight studies [17,18,19,20,21,22,23,24,25,26,27,28,29,30,31,32,33,34,35,36,37,38,39,40,41,42,43,44,45,46,47,48,49,50,51,52,53,54] reported on the relationship between chemical properties and GI, while nine studies reported on the source types of carbohydrates to GI, respectively. Compared to other chemical properties of starchy foods or the source of carbohydrate-based foods, fat content did not reduce GI (SMD: 0.05, 95%CI: −0.16 to 0.27, *p* = 0.312; I^2^ = 93.64%), as was also found in tuber type (SMD: −0.25, 95%CI: −0.93 to 0.43, *p* = 0.233; I^2^ = 79.4%) or fruit type (SMD: 0.25, 95%CI: −0.60 to 1.10, *p* = 0.284; I^2^ = 89.3%).

### 3.6. Strength of Evidence and Publication Bias

The quality of evidence was evaluated by the GRADE system. The level of evidence of RS content was at level A and highly recommended. Phenol content and legume type were at level B and moderately recommended. All evidence profiles for the primary and secondary outcomes are provided in Table 5. For the meta-analysis of RS content to GI food, any publication bias was observed by Begg’s test and Egger’s test (Begg’s, *p* = 0.020; Egger’s, *p* = 0.004) (Figure 5a). For phenol content on GI food, any publication bias was observed by Begg’s test and Egger’s test (Begg’s, *p* = 0.001; Egger’s, *p* = 0.008). For the meta-analysis of legume type on GI food, any publication bias was observed by Begg’s test and Egger’s test (Begg’s, *p* = 0.087; Egger’s, *p* = 0.077).

### 3.7. In Vitro Laboratory Simulation Experiments

Compared to other chemical properties, resistant starch and phenolic content significantly reduced the GI value of starchy foods (respectively, SMD: −2.52, 95%CI: −3.29 to −1.75, *p* < 0.001; SMD: −0.72, 95%CI: −1.26 to−0.17, *p* = 0.005), with heterogeneity (respectively, I^2^ = 84.23%, *p* < 0.001; I^2^ = 73.64%, *p* = 0.005) (Table 2, Table 4, and Table 5). The lowest GI crop type is legumes (SMD: −2.15, 95%CI: −3.45 to −0.85, *p* < 0.001), with heterogeneity (I^2^ = 72.97%, *p* < 0.001) (Figure 7). The heterogeneity among these studies may result from discrepancies in the population and control group. Crop species, i.e., sorghum and red kidney bean had low GI (respectively, SMD: −0.69, 95%CI: −2.33 to 0.96, *p* < 0.001; SMD: −0.39, 95%CI: −2.37 to 1.59, *p* = 0.001), with heterogeneity (respectively, I^2^ = 81.4%, *p* < 0.001; I^2^ = 73.0%, *p* = 0.001) (Table 5 and Figure 7).

### 3.8. In Vivo Experiments

In all in vivo studies, compared to other chemical properties, flavonoid and phenolic content significantly reduced the GI value of starchy foods (respectively, SMD: −0.67, 95%CI: −0.87 to −0.47, *p* < 0.001; SMD: − 0.63, 95%CI: −1.15 to−0.11, *p* = 0.009), with heterogeneity (respectively, I^2^ = 97.34%; I^2^ = 53.54%) (Table 3, Table 6, and Table 7). In in vivo studies with 50 g glucose as the reference food, compared to other chemical properties, phenolic and flavonoid content significantly reduced the GI value of starchy foods (respectively, SMD: −0.63, 95%CI: −1.15 to −0.11, *p* = 0.009; SMD: −0.42, 95%CI: −0.72 to −0.12, *p* = 0.003), with heterogeneity (respectively, I^2^ = 53.54%; I^2^ = 87.50%) (Table 6). The lowest GI crop type is fruit (banana) (SMD: −0.07, 95%CI: −0.95 to 0.80, *p* = 0.433) (Table 3 and Table 7).

## 4. Discussion

### 4.1. Relationship between Food Chemical Properties and GI

This work systematically reviewed the current accessible literature and found that, in general, among the various chemical properties, the presence of resistant starch and phenolic compounds exerted significant impacts on the GI of starchy foods. It is noteworthy that evidence of this finding was consistent with the previous study. Moreover, some chemical properties show an essential impact on the GI beyond those mentioned, i.e., flavonoid, protein, dietary fibre, and amylose, which were negatively correlated with GI. Further, we found that the crop type with the lowest GI value was legumes. Subgroup analysis revealed that significant low-GI crop species were sorghum and red kidney bean, while the subgroup analysis was restricted to trials that compared some crop species. This may relate to a larger quantity of RS and phenolic compounds in the crop species.

No meta-analysis has been conducted on the effect of resistant starch levels on starchy food’s GI. Previous researchers stated that the relationship between resistant starch levels and GI is a negative correlation [19]. Resistant starch is starch that cannot be digested by the small intestine within 120 min after consumption but that the large intestine can ferment. Resistant starch is a linear molecule of α-1,4-d-glucan, which is obtained mainly from the retrogradation of the amylose fraction and has a relatively low molecular weight (1.2 × 10^5^ Da) [74]. The results of a previous meta-analysis showed that resistant starch can reduce blood sugar levels and fasting insulin [9]. The same thing can be seen from the results of the meta-regression between resistant starch levels and effect size (Figure 5b), which has a negative slope (−0.09). The mechanism for decreasing GI is that resistant starch cannot be digested by digestive enzymes due to its compact molecular structure, such that there is no increase in blood sugar levels [74].

Furthermore, meta-analysis has not been conducted on the effect of dietary fibre levels on starchy foods’ GI. Previous researchers stated that the relationship between dietary fibre levels and GI is negatively correlated [6,35]. The same thing can be seen from the result of the meta-regression between dietary fibre content and effect size, which has a negative slope equal to −0.11. The mechanism of action of dietary fibre to reduce GI is by slowing down the rate of digestion of starch and increasing the duration of intestinal transit so that dietary fibre serves as a physical barrier in digestion in the intestine, thus slowing down the interaction between enzymes and substrates. In addition, the degree of viscosity of the dietary fibre is positively related to the extent of the flattening of the postprandial glucose response [75].

Meta-analysis of the effect of fat content on starchy foods’ GI has not been carried out. Previous researchers stated that the relationship of fat content to GI is negatively correlated [76]. The same thing can be seen from the results of the meta-regression between fat content and effect size, which has a negative slope as much as −0.07. The mechanism of GI reduction is that fat slows the rate at which the stomach empties, creates a steric hindrance for the enzyme [77], and interacts with amylose to form a very strong matrix, named amylo-lipid, which digestive enzymes have trouble digesting [75].

In addition, meta-analysis of the effect of protein levels on starchy foods’ GI has not been done. Previous researchers stated that the relationship between protein levels and GI is negatively correlated [55]. The mechanism for decreasing GI is that a protein allegedly stimulates insulin secretion so that blood glucose is not excessive and is under control [78]. However, the meta-regression outcome between protein levels and effect size shows a different result, which has a positive slope of 0.02.

No meta-analysis has been conducted on the effect of phenol levels on starchy foods’ GI. Previous researchers stated that the relationship between phenol levels and GI is negatively correlated [79]. The same can be seen from the result of the meta-regression between phenol levels and effect size, which has a negative slope, as much as −0.0001. The mechanism of GI reduction is that phenol inhibits the α-amylase enzyme and the α-glucosidase enzyme [80].

No meta-analysis has been conducted on the effect of flavonoid levels on starchy foods’ GI. Previous researchers stated that the relationship between flavonoid levels and GI is negatively correlated [66]. The same can be seen from the result of the meta-regression between fat content and effect size, which has a negative slope, as much as −0.0002. The mechanism of GI reduction is that flavonoids inhibit the α-amylase and α-glucosidase enzymes [80].

### 4.2. Comparability of In Vitro Results to Forecast In Vivo Correlations to Chemical Properties and GI

This study used both the in vivo and in vitro methods. This consideration was based on previous studies in which in vitro test results had the same trend as in vivo tests, though in vitro tests tended to have absolute values (overestimation) about 5–25 higher than those of in vivo tests [11,81,82]. That gap was strengthened by the results of the absolute value of SMD chemical properties towards the GI of food, which, in vitro, has a higher value compared to in vivo (Table 4 and Table 6). Other considerations are several theories stating that at least 10 studies must be carried out in a meta-analysis. In the analysis, subgroup analysis and sensitivity analysis are performed. Subgroup analysis was performed based on the variable type of study and reference food. This can provide an overview of two aspects if both methods are used and if only the in vivo method with 50 g glucose reference food os used (Table 6). For in vivo tests, we used and compared only those using 50 g glucose reference food. This is done because the results show that if the reference food is 50 g of white bread or white rice, it is necessary to first convert the GI value with multiplier factors, respectively, 0.77 and 0.69 [83]. If the reference food used is 25 g glucose, different results would be obtained, which would need to be multiplied by 0.67 as a conversion factor [84].

For in vivo food model simulation, we suggest some recommendations. First, our study found that RS and phenolic content had positive effects on the reduction of the GI value of starchy foods. Such a correlation is stable and reliable. Thus, RS and phenolic content should be recommended for the chemical properties of starchy foods determined to affect the GI value. Second, to date, little attention has been paid to the study of the chemical properties determinant that affects the glycemic index of starchy foods and selected low-GI carbohydrates using meta-analysis. The determinant factor discussed in this study can be a meaningful direction for further research. Finally, comprehensive in vivo trials are warranted to validate [17] the positive impact of these findings.

### 4.3. Determinant of Chemical Properties Affecting the GI and Low-GI Carbohydrate Foods

In our study, the determinant effect of RS or phenolic content on GI and legume, as the selected low-GI carbohydrate food, was in accordance with the previous meta-analysis [9]. Nevertheless, differences between our study and the previous analysis should be noted. First, the previous meta-analysis included thirteen trials with the involvement of 428 participants for the RS effect, while for phenolic content to GI, no meta-analysis research was found until now. However, Ramdath et al. (2014) [62] asserted that there was a significant inverse correlation between polyphenol content and the GI of potatoes (r = −0.825; *p* < 0.05; *n* = 4). In the case of the relation between legume and GI, the previous study using meta-analysis included forty trials totaling 253 participants. We included ninety-eight trials, and added subgroup analysis based on chemical properties and the source of starchy foods according to the control group, enabling us to reach a more robust conclusion by eliminating interference factors. Our meta-analysis found that heterogeneity among trials was due mainly to the design of different control group, rather than population. In addition, we assessed the quality of the evidence and the strength of the recommendations. Thus, our work was the latest and most comprehensive one.

Low-GI foods, such as legumes, had the lowest GI compared to other carbohydrate foods because legumes have relatively higher levels of resistant starch and bioactive compounds. Beans’ resistant starch levels were 24.7% [74]. In the cereal group, red sorghum had the lowest GI (Table 5), apparently due to the higher content of bioactive compounds and resistant starch compared to other cereal groups. Red sorghum’s resistant starch levels ranged from 3.34 to 65.36 g/100 g [85]. The phenol compound contained in sorghum is 445–2850 μg/g [86]. In the legumes, red beans had the lowest GI (Table 5), presumably due to the higher content of bioactive compounds and resistant starch compared to other legumes. The resistant starch content of red bean starch is 21.27% [87], while the total phenol content can reach 4871 mg gallic acid equivalents (GAE)/100 g dry weight [88]. Previous studies [89] only determined the glycemic index of various staple carbohydrate-rich foods in the UK diet in five groups, such as breakfast cereals, breads, pastas, and potatoes using ten subjects.

Our study also has limitations. Though this meta-analysis includes high-quality studies, the sample sizes are small. Additionally, there is enough heterogeneity among studies, which could alter the reliability of the results. Trials with higher-quality samples and larger sample sizes are required to confirm the current results.

### 4.4. Critical Review GI as Indicator for Classifying Healthy Foods and the Alternative Concept

Although the concept of GI is widely used in explaining the causes of diabetes, some scientists consider that GI is not accurate enough to explain this. The concept of GI is considered inappropriate to classify a food as healthy or not or to describe its impact on human health. Several aspects of criticism of the GI concept include reproducibility, its impact on physiological effects, and levels and standards of the reference food used [90]. Therefore, it is necessary to use other indicators related to the character of carbohydrates besides digestibility, such as the types of food fibre found in foodstuffs and the levels of bioactive compounds contained therein [91]. Substitutes for the concept of GI proposed by health nutrition researchers include a new method for classifying starch digestion by modeling amylolysis of plant foods using first-order enzyme kinetic principles. This research opens new horizons and supports the relationship between levels of resistant starch, dietary fibre, phenolic, flavonoids, and the value of food GI.

The results evidenced that resistant starch and phenolic content reduce the GI value of starchy foods. As regards in vitro studies, it is well known that resistant starch is negatively correlated to GI, but the results obtained for the phenolic content in this systematic review were not obvious, even though part of phenolics were bound to fibre compounds known to reduce the digestion in the flattening of postprandial glucose response. Moreover, phenols inhibit the α-amylase and α-glucosidase enzymes. Among cereals, sorghum—a gluten free cereal—is the only one that reveals high-resistant starch and low GI. This is a very interesting result due to the fact that gluten-free foods generally are low in fibre and high in GI. This finding could be useful to investigate the potential of sorghum as gluten-free products.

## 5. Conclusions

The present work successfully elucidated that resistant starch, phenolic, flavonoid, protein, and dietary fibre content exerted crucial roles in reducing the glycemic index of starchy foods. Among the starchy foods, sorghum and red kidney bean were identified to have low-GI properties. Sorghum and kidney beans have a low GI because both contain relatively high resistant starch and phenolic compounds. The relationship between levels of resistant starch, phenolic, flavonoids, protein, and fibre to GI was a negatively correlation. Resistant starch causes steric hindrance in the molecular structure of the starch, while phenolic compounds (including flavonoids) are capable of inhibiting the α-amylase and α-glucosidase enzymes. The mode of action of resistant starch in reducing GI is making the enzyme unable to hydrolyze and disrupting the hydrolysis on non-resistant starch (which creates steric hindrance). Nevertheless, microbes ferment the resistant starch in the colon so that the body will not absorb it as glucose. Protein is supposed to stimulate the secretion of insulin so that blood glucose is not excessive and can be controlled. The fibre functions as an inhibitor of physical digestion in the intestine, thereby slowing down the interactions between enzymes with the substrates.

## Figures and Tables

**Figure 1 foods-10-00364-f001:**
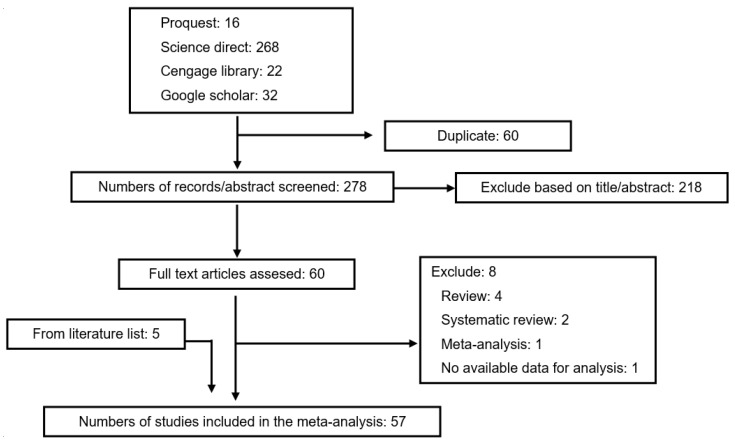
The preferred reporting items for systematic reviews and meta-analyses (PRISMA) flow chart of the literature review process.

**Figure 2 foods-10-00364-f002:**
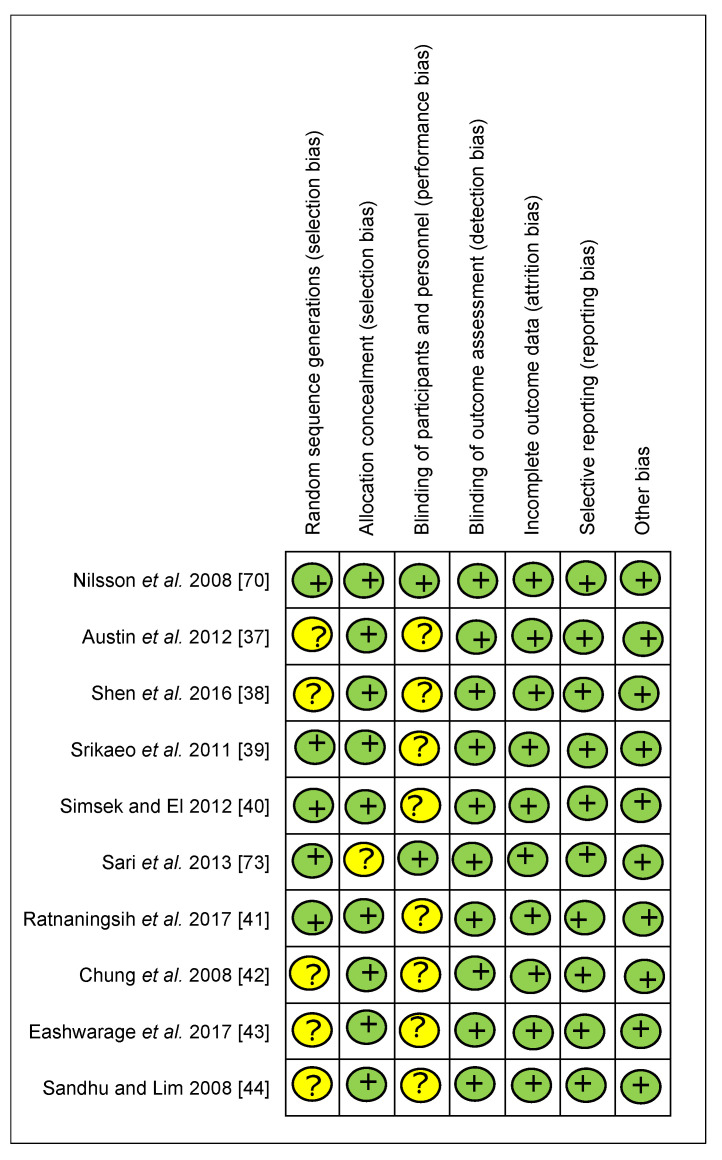
The result of the risk of bias assessment: each risk of bias item for the included studies (green means low risk of bias, yellow means unclear risk of bias, red means high risk of bias).

**Figure 3 foods-10-00364-f003:**
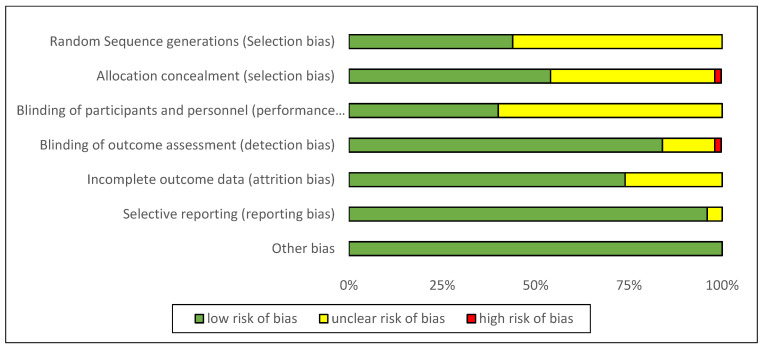
The result of the risk of bias assessment: each risk of bias item is shown as percentages across all included studies.

**Figure 4 foods-10-00364-f004:**
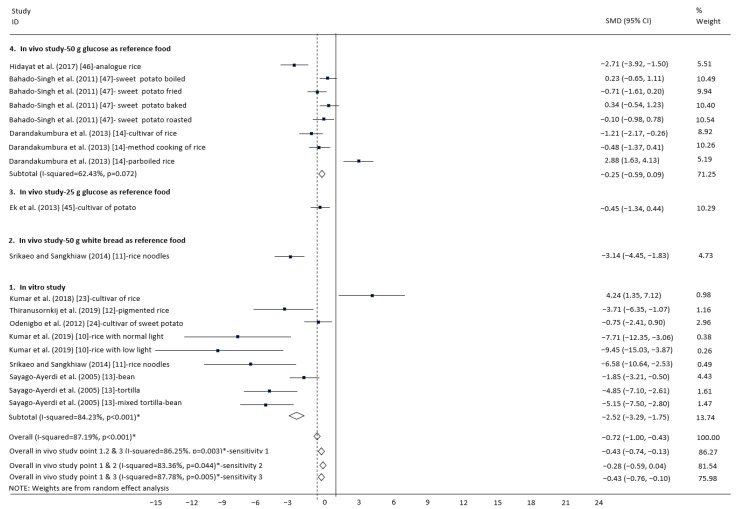
Subgroup analysis of the glycemic index according to the differing resistant starch content in carbohydrate foods. Standardized mean difference (SMD), confidence interval (CI), and point represent the estimated overall effect size (with 95%CIs) for each study. Positive values indicate relatively higher resistant starch content in carbohydrate foods. Negative values indicate relatively lower resistant starch content in carbohydrate foods.

**Figure 5 foods-10-00364-f005:**
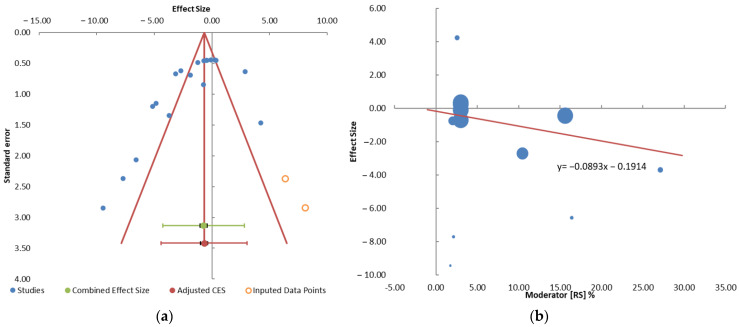
(**a**) The funnel plot of the effect of differing resistant starch content in carbohydrate foods-to-GI; (**b**) the meta-regression between resistant starch content (%) and effect size.

**Figure 6 foods-10-00364-f006:**
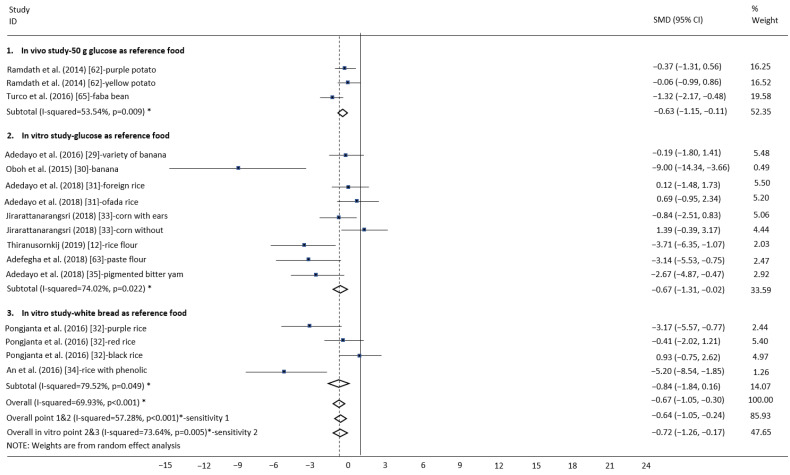
Subgroup analysis of the glycemic index according to the differing phenolic content in carbohydrate foods. Standardized mean difference (SMD), confidence interval (CI), and point represent the estimated overall effect size (with 95%CIs) for each study. Positive values indicate relatively higher phenolic content in carbohydrate foods. Negative values indicate relatively lower phenolic content in carbohydrate foods.

**Figure 7 foods-10-00364-f007:**
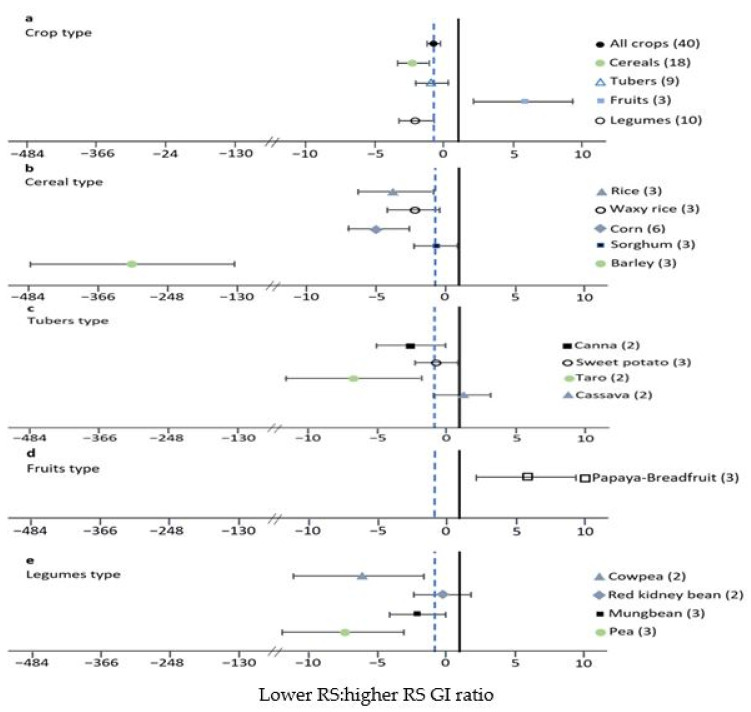
Influence of different crop type (**a**), cereal type (**b**), tuber type (**c**), fruit type (**d**), and legume type (**e**) on lower-RS to higher-RS glycemic index ratio in in vitro studies. Forest plot of cumulative effect size and 95% confidence interval (CI) of different crop types comparing lower RS and higher RS on GI. Bold lines indicate the robust model. Only crop type was represented by at least seventeen observations. Values are mean effect sizes with 95% confidence intervals. The number of studies in each class is written in parentheses. The dotted line shows the cumulative effect size across all classes.

**Table 1 foods-10-00364-t001:** GRADE evidence profile for chemical properties and starchy food sources related to GI.

Quality Assessment							No of Patients/Replicates			Effect		Quality
No of Studies	Design	Risk of Bias	Inconsistency	Indirectness	Imprecision	Other Considerations	In Vitro	In Vivo	Total	Relative (95%CI)	Absolute	
RS content-GI (follow-up 9 days to 7 months; measured with: blood test and in vitro starch hydrolysis; range of scores: 3–5; better indicated by lower values)												
10	randomised trials	no serious risk of bias	no serious inconsistency	no serious indirectness	no serious imprecision	strong association	36	100	136	0.29	SMD −0.72 lower (−1.00 lower to −0.43 higher)	⊕⊕⊕⊕ HIGH
Dietary Fibre content-GI (follow-up 2 days to 6 weeks; measured with: blood test and in vitro starch hydrolysis; range of scores: 2–3; better indicated by lower values)												
15	randomised trials	any serious risk of bias	no serious inconsistency	no serious indirectness	no serious imprecision	strong association	11	210	221	0.22	SMD −0.32 lower (−0.54 lower to −0.10 higher)	⊕⊕⊕⊕ HIGH
Fat content-GI (follow-up 2 days to 6 weeks; measured with: blood test and in vitro starch hydrolysis; range of scores: 2–3; better indicated by lower values)												
15	randomised trials	no serious risk of bias	serious	no serious indirectness	no serious imprecision	strong association	15	216	231	0.21	SMD 0.05 lower (−0.16 lower to 0.26 higher)	⊕⊕⊕⊕ HIGH
Protein content-GI (follow-up 2 days to 6 weeks; measured with: blood test and in vitro starch hydrolysis; range of scores: 2–3; better indicated by lower values)												
15	randomised trials	no serious risk of bias	no serious inconsistency	no serious indirectness	no serious imprecision	strong association	12	226	238	0.20	SMD −0.47 lower (−0.68 lower to −0.27 higher)	⊕⊕⊕⊕ HIGH
Phenol content-GI (follow-up 15 days; measured with: in vitro starch hydrolysis and blood test; range of scores: 5–7; better indicated by lower values)												
12	randomised trials	no serious risk of bias	no serious inconsistency	no serious indirectness	no serious imprecision	strong association	55	18	73	0.38	SMD −0.67 lower (−1.05 lower to −0.30 higher)	⊕⊕⊕O MODERATE
Flavonoid content-GI (follow-up 2 to 5 weeks; measured with: blood test and in vitro starch hydrolysis; range of scores: 4–6; better indicated by lower values)												
10	randomised trials	no serious risk of bias	no serious inconsistency	no serious indirectness	no serious imprecision	strong association	15	305	320	0.20	SMD −0.65 lower (−0.85 lower to −0.45 higher)	⊕⊕⊕⊕ HIGH
Cereals type (follow-up-; measured with: in vitro starch hydrolysis; range of scores: 4–6; better indicated by lower values)												
5	randomised trials	no serious risk of bias	no serious inconsistency	no serious indirectness	serious	reporting bias strong association	18	0	18	1.03	SMD −2.42 lower (−3.45 lower to −1.39 higher)	⊕⊕⊕⊕ HIGH
Tubers type (follow-up 12 to 20 days; measured with: blood test and in vitro starch hydrolysis; range of scores: 3–4; better indicated by lower values)												
6	randomised trials	no serious risk of bias	no serious inconsistency	no serious indirectness	no serious imprecision	strong association	9	14	23	0.68	SMD −0.25 lower (0.43 lower to −0.93 higher)	⊕⊕⊕O MODERATE
Fruits type (follow-up 15 days; measured with: blood test and in vitro starch hydrolysis; range of scores: 3–4; better indicated by lower values)												
2	randomised trials	no serious risk of bias	serious	no serious indirectness	serious	reporting bias strong association	3	10	13	0.85	SMD 0.25 lower (−0.6 lower to 1.1 higher)	⊕⊕⊕⊕ HIGH
Legumes type (follow-up-; measured with: in vitro starch hydrolysis; range of scores: 3–4; better indicated by lower values)												
4	randomised trials	no serious risk of bias	no serious inconsistency	no serious indirectness	no serious imprecision	strong association	10	0	10	1.30	SMD −2.15 lower (−3.5 lower to −0.9 higher)	⊕⊕⊕O MODERATE

**Table 2 foods-10-00364-t002:** Characteristics of in vitro studies included in the systematic review and meta-analysis. RS: resistant starch; GI: glycemic index; M: mean; SD: standard deviation.

Author/Year	Country	Population	No. of Patient/ Replicates	Intervention	Control	Sample/Type	Method/Reference Food	GI Value (M ± SD) Intervention/Control	Follow Up
1. RS Content-GI									
Kumar et al./2018 [23]	India	-	3	Lower RS	Higher RS	Flour rice	In vitro/0.2 g maltose	71.48 ± 0.21 to 69.62 ± 0.45	-
Thiranusornkij et al./2019 [12]	Thailand	-	3	Lower RS	Higher RS	Flour rice	In vitro/100 mg white bread	65.40 ± 4.50 to 87.10 ± 4.85	-
Odenigbo et al./2012 [24]	Canada	-	3	Lower RS	Higher RS	French fries potato	In vitro/50 mg white bread	52.16 ± 2.41 to 53.87 ± 0.89	-
Kumar et al./2019 [10]	India	-	3	Lower RS	Higher RS	Flour rice (2 data)-normal light, low light	In vitro/0.2 g glucose	61.63 ± 1.28 to 72.14 ± 0.86 63.14 ± 0.77 to 74.08 ± 1.06	-
Srikaeo and Sangkhiaw/2014 [11]	Thailand	-	3	Lower RS	Higher RS	Cooked rice noodles	In vitro/100 mg white bread	66.98 ± 2.68 to 83.66 ± 1.02	-
Ayerdi et al./2005 [13]	Mexico	-	6	Lower RS	Higher RS	Cooked bean-corn (3 data)-black beans, tortilla, tortilla-bean	In vitro/1 g white bread	22.00 ± 3.02 to 27.00 ± 1.81 62.00 ± 3.15 to 75.00 ± 1.52 35.00 ± 1.38 to 51.00 ± 3.81	-
2. Dietary Fibre Content-GI									
Vahini et al./2017 [25]	India	-	2	Lower dietary fibre	Higher dietary fibre	Corn flour boiled	In vitro/bread	56.19 ± 1.10 to 82.57 ± 2.25	-
Kim and White/2012 [26]	USA	-	3	Lower dietary fibre	Higher dietary fibre	Oat flour (2 data)-raw, heated	In vitro/100 mg white bread	65.10 ± 1.30 to 61.10 ± 0.90 78.30 ± 1.30 to 85.70 ± 0.70	-
Amin et al./2018 [27]	India	-	3	Lower dietary fibre	Higher dietary fibre	Rice flour by enzymatic hydrolisis	In vitro/50 g white bread	46.10 ± 0.20 to 88.00 ± 0.20	-
3. Fat Content-GI									
Klunkin and Savage/2018 [28]	New Zealand	-	3	Lower fat	Higher fat	Flour purple rice	In vitro/0.5 g glucose	48.56 ± 0.03 to 63.11 ± 0.02	-
Kim and White/2012 [26]	USA	-	3	Lower fat	Higher fat	Oat flour (2 data)-raw, heated	In vitro/100 mg white bread	66.70 ± 1.60 to 64.20 ± 0.40 77.20 ± 0.50 to 82.70 ± 0.90	-
Odenigbo et al./2012 [24]	Canada	-	3	Lower fat	Higher fat	French fries sweet potato	In vitro/50 mg white bread	56.18 ± 0.61 to 54.64 ± 0.71	-
Amin et al./2018 [27]	India	-	3	Lower fat	Higher fat	Rice flour by enzymatic hydrolisis	In vitro/50 g white bread	46.10 ± 0.20 to 88.00 ± 0.20	-
4. Protein Content-GI									
Kim and White/2012 [26]	USA	-	3	Lower protein	Higher protein	Oat flour (2 data)-raw, heated	In vitro/100 mg white bread	64.20 ± 0.40 to 61.10 ± 0.90 77.20 ± 0.50 to 85.70 ± 0.70	-
Odenigbo et al./2012 [24]	Canada	-	3	Lower protein	Higher protein	French fries sweet potato	In vitro/50 mg white bread	52.16 ± 2.41 to 56.18 ± 0.61	-
Amin et al./2018 [27]	India	-	3	Lower protein	Higher protein	Rice flour by enzymatic hydrolisis	In vitro/50 g white bread	46.10 ± 0.20 to 88.00 ± 0.20	-
5. Phenol Content-GI									
Adedayo et al./2016 [29]	Nigeria	-	3	Lower phenol	Higher phenol	Banana fruit	In vitro/50 g glucose	44.95 ± 2.30 to 45.49 ± 2.17	-
Oboh et al./2015 [30]	Nigeria	-	3	Lower phenol	Higher phenol	Banana-breadfruit fruit	In vitro/50 mg glucose	52.78 ± 0.81 to 64.50 ± 1.23	-
Adedayo et al./2018 [31]	Nigeria	-	3	Lower phenol	Higher phenol	Foreign and Ofada Rice (raw-cooked)-2 data	In vitro/50 g glucose	49.00 ± 4.03 to 48.40 ± 3.67 51.80 ± 3.01 to 49.20 ± 2.99	-
Ponjanta et al./2016 [32]	Thailand	-	3	Lower phenol	Higher phenol	Raw rice (white to purple, red, black rice)-3 data	In vitro/50 g white bread	68.50 ± 0.28 to 83.59 ± 5.38 80.40 ± 4.02 to 81.87 ± 0.68 82.79 ± 2.23 to 78.92 ± 4.12	-
Jirarattanarangsri/2018 [33]	Thailand	-	3	Lower phenol	Higher phenol	Cooked corn (purple waxy corn with ears and without ears)-2 data	In vitro/50 g glucose	95.80 ± 0.50 to 96.20 ± 0.20 97.20 ± 0.80 to 96.10 ± 0.40	-
Klunkin and Savage/2018 [28]	New Zealand	-	3	Lower phenol	Higher phenol	Cooked cereal (wheat-purple rice)	In vitro/50 g white bread	48.56 ± 0.03 to 63.11 ± 0.02	-
An et al./2016 [34]	Korea	-	3	Lower phenol	Higher phenol	Rice flour (black rice-phenolic enriched extract)	In vitro/5 g white bread	69.77 ± 1.26 to 79.23 ± 1.63	-
Thiranusornkij et al./2019 [12]	Thailand	-	3	Lower phenol	Higher phenol	Rice flour (Hom Mali-Riceberry flour)	In vitro/50 g glucose	65.40 ± 4.50 to 87.10 ± 4.85	-
Adefegha et al./2018 [63]	Nigeria	-	3	Lower phenol	Higher phenol	Wheat (raw flour-paste flour)	In vitro/50 g glucose	63.15 ± 2.30 to 71.92 ± 2.17	-
Adedayo et al./2018 [35]	Nigeria	-	3	Lower phenol	Higher phenol	Yam (white bitter-yellow bitter)	In vitro/25 mg glucose	56.25 ± 0.81 to 58.95 ± 0.81	-
6. Flavonoid Content-GI									
Adedayo et al./2018 [31]	Nigeria	-	3	Lower flavonoid	Higher flavonoid	Foreign and Ofada Rice (raw-cooked)-2 data	In vitro/50 g glucose	49.00 ± 4.03 to 48.40 ± 3.67 51.80 ± 3.01 to 49.20 ± 2.99	
Klunkin and Savage/2018 [28]	New Zealand	-	3	Lower flavonoid	Higher flavonoid	Cooked cereal (wheat-purple rice)	In vitro/50 g white bread	48.56 ± 0.03 to 63.11 ± 0.02	-
An et al./2016 [34]	Korea	-	3	Lower flavonoid	Higher flavonoid	Rice flour (black rice-phenolic enriched extract)	In vitro/5 g white bread	69.77 ± 1.26 to 79.23 ± 1.63	-
Adefegha et al./2018 [63]	Nigeria	-	3	Lower flavonoid	Higher flavonoid	Wheat (raw flour-paste flour)	In vitro/50 g glucose	63.15 ± 2.30 to 71.92 ± 2.17	-
7. Amylose Content-GI									
Frei et al./2003 [36]	Germany	-	3	Low amylose rice	Medium-high amylose rice	Cooked Rice/Brown Rice-milled	In vitro/50 mg white bread	96.90 ± 4.33 to 68.00 ± 6.41 68.50 ± 5.54 to 87.30 ± 4.68	-
Lower-Higher Category RS [Selected low GI Carbohydrate]									
1. Cereals Category									
Thiranusornkij et al./2019 [12]	Thailand	-	3	Lower RS	Higher RS	Rice	In vitro/50 g glucose	65.40 ± 4.50 to 87.10 ± 4.85	-
Frei et al./2003 [36]	Germany	-	3	Lower RS	Higher RS	Waxy rice	In vitro/50 mg white bread	92.30 ± 8.31 to 109.20 ± 1.56	-
Ayerdi et al./2005 [13]	Mexico	-	6	Lower RS	Higher RS	Corn	In vitro/1 g white bread	62.00 ± 3.15 to 75.00 ± 1.52	-
Austin et al./2012 [37]	USA	-	3	Lower RS	Higher RS	Sorghum	In vitro/50 mg white bread	88.00 ± 5.00 to 92.00 ± 4.30	-
Shen et al./2016 [38]	China	-	3	Lower RS	Higher RS	Barley	In vitro/50 mg white bread	51.65 ± 0.02 to 59.67 ± 0.02	-
2. Tubers Category									
Srikaeo et al./2011 [39]	Thailand	-	2	Lower RS	Higher RS	Canna	In vitro/0.5 g rice noodle 100% amylose	88.00 ± 1.80 to 97.00 ± 2.20	-
Odenigbo et al./2012 [24]	Canada	-	3	Lower RS	Higher RS	Sweet potato	In vitro/50 mg white bread	52.16 ± 2.41 to 53.87 ± 0.89	-
Simsek and El/2012 [40]	Turkey	-	2	Lower RS	Higher RS	Taro	In vitro/50 mg white bread	74.10 ± 1.30 to 86.60 ± 0.80	-
Srikaeo et al./2011 [39]	Thailand	-	2	Lower RS	Higher RS	Cassava	In vitro/0.5 g rice noodle 100% amylose	109.00 ± 1.20 to 105.00 ± 2.40	-
3. Fruits Category									
Oboh et al./2015 [30]	Nigeria	-	3	Lower RS	Higher RS	Papaya-breadfruit	In vitro/50 mg glucose	64.50 ± 1.23 to 55.29 ± 1.33	-
4. Legumes Category									
Ratnaningsih et al./2017 [41]	Indonesia	-	2	Lower RS	Higher RS	Cowpea	In vitro/100 mg white bread	45.46 ± 0.23 to 47.74 ± 0.19	-
Chung et al./2008 [42]	Canada	-	2	Lower RS	Higher RS	Red kidney bean	In vitro/50 mg white bread	12.00 ± 0.10 to 12.20 ± 0.40	-
Eashwarage et al./2017 [43]	Sri Lanka	-	3	Lower RS	Higher RS	Mung bean	In vitro/1 g white bread	41.54 ± 0.25 to 42.05 ± 0.09	-
Sandhu and Lim/2008 [44]	South Korea	-	3	Lower RS	Higher RS	Pea	In vitro/50 g white bread	44.20 ± 0.60 to 48.90 ± 0.40	-

**Table 3 foods-10-00364-t003:** Characteristics of in vivo studies included in the systematic review and meta-analysis. RS: resistant starch; GI: glycemic index; M: mean; SD: standard deviation.

Author/Year	Country	Population	No. of Patient/Replicates	Intervention	Control	Sample/Type	Method/Reference Food	GI Value (M ± SD) Intervention/Control	Follow Up
1. RS Content-GI									
Ek et al./2013 [45]	Australia	Adults (healthy)	10, 27/10 (12 males, 17 females)	Lower RS	Higher RS	Boiled potato (4′)	In vivo/50 mg glucose	82.00 ± 9.49 to 93.00 ± 31.62	12–20 days
Hidayat et al./2017 [46]	Indonesia	Adults (healthy)	10, 10/10	Lower RS	Higher RS	Corn based-rice analogues	In vivo/50 g glucose	34.79 ± 2.11 to 40.77 ± 2.12	12 days
Singh et al./2011 [47]	Jamaica	Adults (healthy)	10/10–5 males, 5 females	Lower RS	Higher RS	Cooked sweet potato (4 data)-boiled, fried, baked, roasted	In vivo/50 g glucose	49.00 ± 12.65 to 46.00 ± 12.65 68.00 ± 9.49 to 75.00 ± 9.49 91.00 ± 9.49 to 87.00 ± 12.65 89.00 ± 9.49 to 90.00 ± 9.49	7 months
Srikaeo and Sangkhiaw/2014 [11]	Thailand	Adults (healthy)	10/10 -5 males, 5 females	Lower RS	Higher RS	Cooked rice noodles	In vivo/50 g bread	51.84 ± 1.95 to 63.62 ± 4.69	9 days
Darandakumbura et al./2013 [14]	Srilanka	Adults (healthy)	10/10–5 males, 5 females	Lower RS	Higher RS	Cooked rice (3 data)-cultivar, cooking method, parboiled	In vivo/50 g glucose	68.00 ± 2.00 to 70.00 ± 1.00 71.00 ± 2.00 to 72.00 ± 2.00 71.00 ± 1.00 to 68.00 ± 1.00	1.5 months (42 days)
2. Dietary Fibre Content-GI									
Oboh and Ogbebor/2010 [48]	Nigeria	Adults (healthy)	10–5 males, 5 females	Lower dietary fibre	Higher dietary fibre	Maize cooking method	In vivo/50 g glucose	38.89 ± 3.13 to 21.32 ± 4.40	15 days
Eli-Cophie et al./2017 [49]	Ghana	Adults (healthy)	10/10–8 males, 2 females	Lower dietary fibre	Higher dietary fibre	Cassava Corn-Fermented Corn (local food in Ghana)	In vivo/50 g glucose	41.00 ± 21.50 to 73.00 ± 15.81	21 days
Mlotha et al./2016 [50]	Malawi	Adults (healthy)	11/11–6 males, 5 females	Lower dietary fibre	Higher dietary fibre	Cooked dehulled-whole maize (porridges)	In vivo/50 g glucose	106.72 ± 47.83 to 74.90 ± 46.22	21 days
Shobana et al./2007 [51]	India	Adults (healthy)	8/8–5 males, 3 females	Lower dietary fibre	Higher dietary fibre	Cooked expanded rice-wheat (food formulation)	In vivo/50 g white bread	55.40 ± 9.00 to 105.00 ± 6.00	25 days
Cavallerro et al./2002 [52]	Italy	Adults (healthy)	8/8–3 males, 5 females	Lower dietary fibre	Higher dietary fibre	100% Bread Wheat-20% Water Extracted Fraction Barley	In vivo/50 g glucose	69.67 ± 20.34 to 89.49 ± 35.10	15 days
Meija et al./2019 [53]	Latvia-Norway	Adults (healthy)	21/18 9–5 male, 4 female	Lower dietary fibre	Higher dietary fibre	Cooked wheat-triticale	In vivo/50 g glucose	53.85 ± 4.86 to 75.76 ± 3.26	5 days
Wolever et al./2018 [54]	Canada	Adults (healthy)	51/40–24 male, 16 female	Lower dietary fibre	Higher dietary fibre	Cooked rice cereal-oat	In vivo/50 g rice cereal	53.85 ± 4.86 to 75.76 ± 3.26	2–6 weeks
Senavirathna et al./2014 [55]	Sri Lanka	Adults (healthy)	10/10	Lower dietary fibre	Higher dietary fibre	Cooked underutilized tuber	In vivo/50 g white bread	82.00 ± 25.30 to 64.00 ± 28.46	≈15 days
Hettiaratchi et al./2011 [56]	Sri Lanka	Adults (healthy)	10	Lower dietary fibre	Higher dietary fibre	Banana fruit	In vivo/50 g white bread	61.00 ± 15.81 to 67.00 ± 22.14	≈15 days
Kouassi et al./2009 [57]	Cote d’Ivoire	Adults (healthy)	10–10 males	Lower dietary fibre	Higher dietary fibre	Cooked yam (2 data)-oven, boiled	In vivo/50 g glucose	66.00 ± 34.64 to 53.00 ± 22.52 54.00 ± 32.91 to 60.00 ± 31.18	≈27 days
Ramdath et al./2004 [58]	Carribean	Adults (healthy)	8–10-4 males, 4–6 females	Lower dietary fibre	Higher dietary fibre	Cooked staple in Carribean (Tannia-green banana)	In vivo/50 g white bread	61.00 ± 15.81 to 67.00 ± 22.14	≈1 months
Omage and Omage/2018 [7]	Nigeria	Adults (healthy)	240/80–40 males, 40 females	Lower dietary fibre	Higher dietary fibre	Cooked mixed rice bean-rice plantain	In vivo/50 g glucose	89.74 ± 9.84 to 86.60 ± 49.82	2 days
3. Fat Content-GI									
Oboh and Ogbebor/2010 [48]	Nigeria	Adults (healthy)	10–5 males, 5 females	Lower fat	Higher fat	Cooked maize	In vivo/50 g glucose	38.89 ± 3.13 to 21.32 ± 4.40	≈9 days
Mlotha et al./2016 [50]	Malawi	Adults (healthy)	11–6 males, 5 females	Lower fat	Higher fat	Cooked maize	In vivo/50 g glucose	106.72 ± 47.83 to 74.90 ± 46.22	21 days
Shobana et al./2007 [51]	India	Adults (healthy)	8–5 males, 3 females	Lower fat	Higher fat	Cooked rice (popped rice)	In vivo/50 g white bread	109.00 ± 8.00 to 55.40 ± 9.00	25 days
Cavallerro et al./2002 [52]	Italy	Adults (healthy)	8–3 males, 5 females	Lower fat	Higher fat	Bread (20% Water-Extracted Fraction Barley-50% Sieved Fraction Barley)	In vivo/50 g glucose	74.46 ± 43.78 to 69.67 ± 20.34	15 days
Wolever et al./2018 [54]	Canada	Adults (healthy)	40–24 male, 16 female	Lower fat	Higher fat	Cooked rice cereal-oat	In vivo/50 g rice cereal	62.98 ± 9.07 to 100.00 ± 0.00	2–6 weeks
Senavirathna et al./2014 [55]	Sri Lanka	Adults (healthy)	10	Lower fat	Higher fat	Cooked underutilized tuber	In vivo/50 g white bread	110.00 ± 25.30 to 69.00 ± 12.65	≈15 days
Shobana et al./2012 [64]	India	Adults (healthy)	23/23–12 males, 11 females	Lower fat	Higher fat	Cooked rice	In vivo/50 g glucose	77.00 ± 19.18 to 72.00 ± 21.58	8 days
Hettiaratchi et al./2011 [56]	Sri Lanka	Adults (healthy)	10	Lower fat	Higher fat	Banana fruit	In vivo/50 g white bread	69.00 ± 28.46 to 61.00 ± 18.97	≈15 days
Kouassi et al./2009 [57]	Cote d’Ivoire	Adults (healthy)	10–10 males	Lower fat	Higher fat	Cooked yam (2 data)-oven, boiled	In vivo/50 g glucose	70.00 ± 31.18 to 53.00 ± 22.52 67.00 ± 27.71 to 51.00 ± 22.52	≈27 days
Ramdath et al./2004 [58]	Carribean	Adults (healthy)	8–10	Lower fat	Higher fat	Cooked staple in Carribean (white yam-dasheen)	In vivo/50 g white bread	88.00 ± 28.46 to 109.00 ± 44.27	≈1 months
Omage and Omage/2018 [7]	Nigeria	Adults (healthy)	80–40 males, 40 females	Lower fat	Higher fat	Cooked mixed rice bean-rice plantain	In vivo/50 g glucose	86.93 ± 56.71 to 86.60 ± 49.82	2 days
4. Protein Content-GI									
Mlotha et al./2016 [50]	Malawi	Adults (healthy)	11–6 males, 5 females	Lower protein	Higher protein	Cooked dehulled-whole maize (porridges)	In vivo/50 g glucose	106.72 ± 47.83 to 121.97 ± 38.99	21 days
Shobana et al./2007 [51]	India	Adults (healthy)	8–5 males, 3 females	Lower protein	Higher protein	Cooked millet-wheat (food formulation)	In vivo/50 g white bread	55.40 ± 9.00 to 93.40 ± 7.00	25 days
Cavallerro et al./2002 [52]	Italy	Adults (healthy))	8–3 males, 5 females	Lower protein	Higher protein	Bread (50% Barley Flour-100% Bread Wheat)	In vivo/50 g glucose	89.49 ± 35.10 to 85.42 ± 38.81	15 days
Wolever et al./2018 [54]	Canada	Adults (healthy)	40–24 male, 16 female	Lower protein	Higher protein	Cooked rice cereal-oat	In vivo/50 g rice cereal	62.98 ± 9.07 to 100.00 ± 0.00	2–6 weeks
Senavirathna et al./2014 [55]	Sri Lanka	Adults (healthy)	10	Lower protein	Higher protein	Cooked underutilized tuber	In vivo/50 g white bread	64.00 ± 28.46 to 69.00 ± 12.65	≈15 days
Shobana et al./2012 [64]	India	Adults (healthy)	23–12 males, 11 females	Lower protein	Higher protein	Cooked rice	In vivo/50 g glucose	70.20 ± 17.26 to 77.00 ± 19.18	8 days
Kouassi et al./2009 [57]	Cote d’Ivoire	Adults (healthy)	10–10 males	Lower protein	Higher protein	Cooked yam (2 data)-oven, boiled	In vivo/50 g glucose	53.00 ± 22.52 to 56.00 ± 20.78 60.00 ± 31.18 to 54.00 ± 32.91	≈27 days
Ramdath et al./2004 [58]	Carribean	Adults (healthy)	8–10	Lower protein	Higher protein	Cooked staple in Carribean (tannia-green banana)	In vivo/50 g white bread	88.00 ± 28.46 to 109.00 ± 44.27	≈1 months
Omage and Omage/2018 [7]	Nigeria	Adults (healthy)	80–40 males, 40 females	Lower protein	Higher protein	Cooked mixed rice bean-rice plantain	In vivo/50 g glucose	86.93 ± 56.71 to 89.74 ± 9.84	2 days
Oboh et al./2010 [59]	Nigeria	Adults (healthy)	5	Lower protein	Higher protein	Cooked legumes (Cowpea black white-Pigeonpea brown)	In vivo/50 g glucose	24.00 ± 22.36 to 30.00 ± 24.60	≈24 days
Araya et al./2003 [60]	Chile	Adults (healthy)	10 males	Lower protein	Higher protein	Cooked legumes (bean-lentil)	In vivo/50 g white bread	49.30 ± 29.50 to 76.80 ± 43.40	≈3–6 weeks
Dhaheri et al./2017 [61]	United Arab Emirates	Adults (healthy)	37/15- 6 males, 9 females	Lower protein	Higher protein	Emirate cuisine (balalet-chami)	In vivo/50 g glucose	60.00 ± 36.00 to 63.00 ± 19.36	≈2 months
5. Phenol Content-GI									
Ramdath et al./2014 [62]	Canada	Caucasian Adults (healthy)	9/9–3 males, 6 females	Lower phenol	Higher phenol	Pigmented potatoes (white-purple; red-yellow)-2 data	In vivo/50 g glucose	77.00 ± 27.00 to 93.00 ± 51.00 78.00 ± 42.00 to 81.00 ± 48.00	≈15 days
Turco et al./2016 [65]	Italy	Adults (healthy)	13- 2 males, 11 females	Lower phenol	Higher phenol	Cooked pasta wheat-bean	In vivo/50 g glucose	40.00 ± 16.22 to 72.00 ± 28.84	≈15 days
6. Flavonoid Content-GI									
Turco et al./2016 [65]	Italy	Adults (healthy)	13/13- 2 males, 11 females	Lower flavonoid	Higher flavonoid	Cooked pasta wheat-bean	In vivo/50 g glucose	40.00 ± 16.22 to 72.00 ± 28.84	≈15 days
Abubakar et al./2018 [66]	Malaysia	Rats weighing 160–200 g	60 males Sprague-Dawley rats	Lower flavonoid	Higher flavonoid	Rice flour (germinated brown rice, brown rice, white rice)-3 data	In vivo/500 mg glucose	67.60 ± 2.27 to 64.30 ± 3.51 67.70 ± 2.04 to 81.80 ± 2.70 65.60 ± 3.51 to 84.70 ± 2.81	≈5 weeks
Raghuvanshi et al./2017 [67]	India	Adults (healthy)	10	Lower flavonoid	Higher flavonoid	Cooked rice (red-white rice)	In vivo/50 g glucose	71.70 ± 2.63 to 63.15 ± 0.91	≈15 days
Meera et al./2019 [68]	India	Adults (healthy)	12	Lower flavonoid	Higher flavonoid	Cooked rice (red-white rice)	In vivo/50 g glucose	47.19 ± 11.09 to 69.74 ± 15.59 61.69 ± 13.86 to 56.27 ± 14.90	≈15 days
Yulianto et al./2018 [69]	Indonesia	Adults (healthy)	12	Lower flavonoid	Higher flavonoid	Cooked rice (cinnamon bark, pandan leaf, bay leaf extract)-3 data	In vivo/50 g glucose	29.00 ± 8.65 to 32.00 ± 8.70 33.00 ± 13.70 to 40.00 ± 13.92 31.00 ± 8.06 to 37.00 ± 10.69	≈1 month
Nilsson et al./2008 [70]	Swedish	Adults (healthy)	12/12–7 males, 5 females	Lower flavonoid	Higher flavonoid	Cooked cereal (rye-wheat; oat-barley)-2 data	In vivo/50 g glucose	79.00 ± 45.03 to 73.00 ± 65.82 49.00 ± 24.25 to 85.00 ± 45.03	≈28 days
7. Amylose Content-GI									
Nounmusig et al./2018 [71]	Thailand	Adults (healthy)	22 22/9	Low amylose rice	Medium-high amylose rice	Cooked rice/White rice	In vivo/50 g glucose	90.70 ± 36.00 to 66.10 ± 33.00 90.70 ± 36.00 to 63.80 ± 37.50 90.70 ± 36.00 to 66.20 ± 24.00 90.70 ± 36.00 to 54.60 ± 19.50 90.70 ± 36.00 to 48.10 ± 18.60	3 weeks
Trinidad et al./2014 [72]	Philippines	Adults (healthy)	12 12/12	Low amylose rice	Medium-high amylose rice	Cooked rice/White rice-milled (3 data) and brown rice (3 data)	In vivo/50 g glucose	69.00 ± 13.86 to 59.00 ± 10.39 85.00 ± 10.39 to 59.00 ± 10.39 94.00 ± 17.32 to 59.00 ± 10.39 61.00 ± 10.39 to 57.00 ± 10.39 69.00 ± 13.86 to 57.00 ± 10.39 77.00 ± 17.32 to 57.00 ± 10.39	≈27 days
Trinidad et al./2013 [8]	Philippines	Adults (healthy)	10 10/10	Low amylose rice	Medium-high amylose rice	Cooked Rice/White Rice-milled (3 data) and brown rice (1 data)	In vivo/50 g glucose	59.00 ± 12.65 to 50.00 ± 9.49 75.00 ± 12.65 to 63.00 ± 9.49 70.00 ± 12.65 to 57.00 ± 9.49 55.00 ± 6.32 to 51.00 ± 3.16	≈1 month
Lower-Higher Category RS [Selected low GI Carbohydrate]									
1. Tubers Category									
Ek et al./2013 [45]	Australia	Adults (healthy)	10	Lower RS	Higher RS	Potato	In vivo/50 mg glucose	82.00 ± 9.49 to 93.00 ± 31.62	12–20 days
Sari et al./2013 [73]	Indonesia	Rats Wistar 200–220 g age 2–3 months	4	Lower RS	Higher RS	Underutilized tubers	In vivo/50 g glucose	22.40 ± 0.00 to 20.60 ± 0.49	≈12 days
2. Fruits Category									
Hettiaratchi et al./2011 [56]	Sri Lanka	Adults (healthy)	10	Lower dietary fibre	Higher dietary fibre	Banana	In vivo/50 g white bread	67.00 ± 22.14 to 69.00 ± 28.46	≈15 days

Note: Explanation in intervention and control column stated that lower fat and higher fat or lower protein and higher protein; for example, Wolever et al. (2018) [54] and Ramdath et al. (2004) [58]. This means that the data used in the control are the data that have a proximate or bioactive compound analysis result (like fat, protein, phenolic, flavonoid, and others) that is higher than the data used in the intervention, such as in, e.g., Ramdath et al. (2004), who compared tannia and green banana. Green banana had higher protein content (2.7 g per serving food for glycemic index (GI) test) than tannia (2.6 g per serving food for GI test). The GI value of green banana is 109.00 ± 44.27, while that of tannia is 88.00 ± 28.46. Therefore, we input the GI value of green banana as the control data and the GI value of tannia as intervention data.

**Table 4 foods-10-00364-t004:** Chemical properties of starchy foods determined to affect the GI in in vitro studies.

No.	Chemical Properties	*n*	I^2^	*p*-Value	Significancy One Tailed α 5%	d+	% (Δ to 1 Value)	Determinant Order
1	RS content	9	84.23	<0.001	significant	−2.52	352	1
2	Dietary fibre content	3	91.20	0.348	not significant	−0.37	137	4
3	Fat content	3	86.39	0.078	not significant	0.91	9	6
4	Protein content	3	90.80	0.251	not significant	−0.51	151	3
5	Phenolic content	13	73.64	0.005	significant	−0.72	172	2
6	Flavonoid content	3	71.74	0.309	not significant	−0.26	126	5

**Table 5 foods-10-00364-t005:** Selected low-GI carbohydrate foods using meta-analysis in in vitro studies.

No.	Item	*n*	I^2^ (%)	*p*-Value	Significancy One Tailed α 5%	% Weight	CI	GI Value from Study [4,21,22]	Selected Low GI Carbohydrate Foods
Crop Type									
1	All crops	40	86.35	<0.001	significant	100.00	0.44	-	-
	Cereals					11.67	1.03	56.44	-
	Tubers					18.58	0.68	55.97	-
	Fruits					42.79	0.85	50.76	-
	Legumes					26.96	1.30	41.94	√
Cereal Type									
2	Rice	18	81.41	<0.001	significant	15.13	2.64	67.9	-
	Waxy rice					25.12	2.05	79	-
	Corn					20.88	2.25	48	-
	Sorghum					38.87	1.65	32	√
	Barley					0.00	175	40	-
Tuber Type									
3	Canna	9	51.59	0.076	not significant	18.63	2.64	19.87	-
	Sweet potato					47.45	1.66	70	
	Taro					5.24	4.99	63.1	
	Cassava					28.68	2.13	78.7	
Fruit Type									
4	Papaya-Breadfruit	3	-	<0.001	significant	100.00	3.63	64.5	-
Legume Type									
5	Cowpea	10	72.97	<0.001	significant	7.57	4.71	41	-
	Red bean					42.84	1.98	26	√
	Mungbean					41.20	2.02	76	-
	Pea					8.39	4.47	35	-

**Table 6 foods-10-00364-t006:** Chemical properties of starchy foods determined to affect the GI in in vivo studies with 50 g glucose as the reference food.

No.	Chemical properties	*n*	I^2^	*p* Value	Significancy One Tailed α 5%	d+	% (Δ to 1 Value)	Determinant Order
1	RS content	8	62.43	0.072	not significant	−0.25	125	3
2	Dietary fibre content	8	89.55	0.041	significant	−0.23	123	4
3	Fat content	7	79.06	0.033	significant	0.23	77	6
4	Protein content	8	0.00	<0.001	significant	−0.13	113	5
5	Phenolic content	3	53.54	0.009	significant	−0.63	163	1
6	Flavonoid content	7	87.50	0.003	significant	−0.42	142	2

**Table 7 foods-10-00364-t007:** Selected low-GI carbohydrate foods using meta-analysis in in vivo studies.

No	Item	*n*	I^2^ (%)	*p*-Value	Significancy One Tailed α 5%	% Weight	CI	GI Value from Study [4,21,22]	Selected Low GI Carbohydrate Foods
Crop Type									
1	All crops	24	86.35	0.413	not significant	100.00	0.53	-	-
	Tubers					61.35	0.68	56.44	-
	Fruits					38.65	0.85	50.76	-
Tuber Type									
2	Potato	14	59.76	0.440	not significant	89.56	0.89	82	-
	Underutilized tuber					10.44	2.60	-	-
Fruit Type									
3	Banana	10	-	0.433	not significant	100.00	0.88	52.78	-

## Data Availability

Not applicable.

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
