# Peer review of "Evaluation of Various Starchy Foods: A Systematic Review and Meta-Analysis on Chemical Properties Affecting the Glycemic Index Values Based on In Vitro and In Vivo Experiments"

_foods, 2021, doi:10.3390/foods10020364_

Round 1

Reviewer 1 Report

After reading the manuscript, I am impressed by the work done and I have editorial comments, which I present below:

-title, abstract and introduction is well written.

-in most tables and figures (e.g. table 1, 2, 3 and figures 2, 4, 6, 7) the font is too small and it makes it difficult to read the contents of the tables

-the caption under figure 5 seems incomplete,

Please verify the conclusions as they are too general. Moreover i think that the conclusions should be broadened to include the term of how individual groups of nutrients affect GI especially the aim of work is: "At present, quantitative studies pertaining to the linkages between chemical properties, the GI of starchy foods, and selected low-GI carbohydrate-based foods are scarce."

Best regards

Author Response

Reviewers comments

Response

Information

Reviewer 1

-title, abstract and introduction is well written.

-in most tables and figures (e.g. table 1, 2, 3 and figures 2, 4, 6, 7) the font is too small and it makes it difficult to read the contents of the tables

-the caption under figure 5 seems incomplete,

Please verify the conclusions as they are too general. Moreover i think that the conclusions should be broadened to include the term of how individual groups of nutrients affect GI especially the aim of work is: "At present, quantitative studies pertaining to the linkages between chemical propertiesthe GI of starchy foods, and selected low-GI carbohydrate-based foods are scarce."

Thank you for your comment.

We have revised it by increase the word size for the tables and by increase figure size for the figures.

We have revised it to: “Figure 5. (a) The funnel plot of the effect of differing resistant starch content in carbohydrate foods-to-GI; (b) The meta-regression between resistant starch content (%) and effect size.”

We have added in the conclusion an explanation with the phrase: “Sorghum and kidney beans have a low GI because both contain relatively high resistant starch and phenolic compounds. The relationship between levels of resistant starch, phenolic, flavonoids, protein, and fiber to GI was negatively correlated. Resistant starch causes steric hindrance in the molecular structure of the starch, while phenolic compounds (including flavonoids) is capable to inhibit the a-amylase and a-glucosidase enzymes. The mode of action of resistant starch in reducing GI is making enzyme unable to hydrolyze and disrupting the hydrolysis on non-resistant starch (creates steric hindrance). Nevertheless, microbes ferment the resistant starch in the colon so that the body will not absorb it as glucose. Protein is supposed to stimulate the secretion of insulin so that blood glucose is not excessive and can be controlled. The fiber functions as an inhibitor of physical digestion in the intestine thereby slowing down the interactions between enzymes with the substrates.”

-

-

Line 209-210

Line 426-437

Reviewer 2 Report

This systematic review and meta- analysis highlighted interesting results. The results  evidenced that resistant starch  and phenolic content reduce the GI value of starchy foods. As regards in vitro studies it is well known that resistant starch is negatively correlated to GI but the results obtained for the phenolic content in this sistematic review are not obvious though part of phenolics are bound to fibre compounds known to reduce the digestion in the intestine flattening the post prandial glucose response. Moreover, as stated in the manuscript, phenols inhibit the alpha-amylase and amyloglucosidase enzymes. Among cereals sorghum, a gluten free cereal, is the one revealing high resistant starch and low GI. This is a very interesting result due to the fact that gluten free foods generally are low in fibre and high in GI. This finding could be usefull to investivate the potential of sorghum gluten free products. This systematic review, analysing many litterature data, give to researchers a good overview about the chemical properties affecting the GI. No observations about the statistical methods and programs used for the realization of the meta-analysis are present due to my limited experience on this though to my opinion all the criteria were correctly considered.

This is a very interesting article.

minor revisions

line 37: delete hyphen between dia-betes

Line 42:delete hyphen between con-sumes

line 43: delete hyphen between sys-tem

line 49: delete hyphen between responsi-ble

figure 2: parts of the figure are out of focus, please try also to increase the words size.

Figure 4,6 and 7: try to increase the figure size or the words and numbers size

Author Response

Reviewer 2

This systematic review and meta-analysis highlighted interesting results. The results evidenced that resistant starch and phenolic content reduce the GI value of starchy foods. As regards in vitro studies it is well known that resistant starch is negatively correlated to GI but the results obtained for the phenolic content in this sistematic review are not obvious though part of phenolics are bound to fibre compounds known to reduce the digestion in the intestine flattening the post prandial glucose response. Moreover, as stated in the manuscript, phenols inhibit the alpha-amylase and amyloglucosidase enzymes. Among cereals sorghum, a gluten free cereal, is the one revealing high resistant starch and low GI. This is a very interesting result due to the fact that gluten free foods generally are low in fibre and high in GI. This finding could be usefull to investivate the potential of sorghum gluten free products. This systematic review, analysing many litterature data, give to researchers a good overview about the chemical properties affecting the GI. No observations about the statistical methods and programs used for the realization of the meta-analysis are present due to my limited experience on this though to my opinion all the criteria were correctly considered.

This is a very interesting article.

minor revisions

Line 37: delete hyphen between dia-betes

Line 42:delete hyphen between con-sumes

Line 43: delete hyphen between sys-tem

Line 49: delete hyphen between responsi-ble

Figure 2: parts of the figure are out of focus, please try also to increase the words size.

Figure 4,6 and 7: try to increase the figure size or the words and numbers size

We think that the results obtained for the phenolic content in this systematic review are clear as in in vitro studies in line 253-254

We have added “We used weighted analysis using Hedges’ d (Standard Mean Difference/ SMD) for statistical methods. The data extracted from selected journals are mean, standard deviation or standard error, and the number of replicate experiments.” And

We used Meta-Essentials tools for meta-analysis process” to explain program used.

We had accomodated input from reviewer in the manuscript with phrase: “The results evidenced that resistant starch and phenolic content reduce the GI value of starchy foods. As regards in vitro studies it is well known that resistant starch is negatively correlated to GI but the results obtained for the phenolic content in this systematic review were not obvious though part of phenolics were bound to fibre compounds known to reduce the digestion in the flattening of postprandial glucose response. Moreover, phenols inhibit the a-amylase and a-glucosidase enzymes. Among cereals, sorghum -gluten free cereal- is the only one which reveal high resistant starch and low GI. This is a very interesting result due to the fact that gluten free foods generally are low in fibre and high in GI. This finding could be useful to investigate the potential of sorghum as gluten free products

Thank you for your comment.

Thank you for your comment.

We have revised it by deleted the hyphen.

We have revised it by deleted the hyphen.

We have revised it by deleted the hyphen.

We have revised it by deleted the hyphen.

We have revised it by increased the word size.

We have revised it by increased the figures size.

-

Line 122-124

Line 128-129

Line 412-421

-

-

-

-

-

-

-

-

Reviewer 3 Report

This is an important topic, but the present submission is unsuitable in its present form.

  1. Although the use of GI has become widespread, a critical review should spend some time discussing major criticisms of this, which in fact claim, with some legitimacy, that it is invalid for human health, e.g. DeVries, J. W., Glycemic Index: The Analytical Perspective, Cereal Foods World 2007, 52, 45-9, and subsequent papers. Also a critical review should examine replacements for this, as discussed in Edwards, C. H.; Warren, F. J.; Milligan, P. J.; Butterworth, P. J.; Ellis, P. R., A novel method for classifying starch digestion by modelling the amylolysis of plant foods using first-order enzyme kinetic principles, Food & Function 2014, 5, 2751-2758, and preceding papers by this group. Missing these renders the present paper no more than the type of uncritical literature review that can be done purely by a computer literature search.
  2. The manuscript has not been properly proof-read by the authors, as evidenced by the many extraneous hyphens scattered throughout.
  3. The Introduction states that “quantitative studies pertaining to the linkages between chemical properties, the GI of starchy foods, and selected low-GI carbohydrate-based foods are scarce.” This is  wrong: there is a huge number of such studies in the literature.

Author Response

Reviewer 3

This is an important topic, but the present submission is unsuitable in its present form.

  1. Although the use of GI has become widespread, a critical review should spend some time discussing major criticisms of this, which in fact claim, with some legitimacy, that it is invalid for human health, e.g. DeVries, J. W., Glycemic Index: The Analytical Perspective, Cereal Foods World 2007, 52, 45-9, and subsequent papers. Also a critical review should examine replacements for this, as discussed in Edwards, C. H.; Warren, F. J.; Milligan, P. J.; Butterworth, P. J.; Ellis, P. R., A novel method for classifying starch digestion by modelling the amylolysis of plant foods using first-order enzyme kinetic principles, Food & Function 2014, 5, 2751-2758, and preceding papers by this group. Missing these renders the present paper no more than the type of uncritical literature review that can be done purely by a computer literature search.

  1. The manuscript has not been properly proof-read by the authors, as evidenced by the many extraneous hyphens scattered throughout.

  1. The Introduction states that “quantitative studies pertaining to the linkages between chemical properties, the GI of starchy foods, and selected low-GI carbohydrate-based foods are scarce.” This is  wrong: there is a huge number of such studies in the literature.

We accomodated the suggestion by added subheading and one paragraph:” 4.4. Critical Review GI as Indicator for Classifying Healthy Foods and The Alternative Concept.

Although the concept of GI is widely used in explaining the causes of diabetes, some scientist consider that GI is not accurate enough to explain this. The concept of GI is considered inappropriate to classify a food as healthy or not or to describe its impact on human health. Several aspects of criticism of the GI concept include reproducibility, its impact on physiological effects, and levels and standards of the reference food used [86]. So it is necessary to use other indicators related to the character of carbohydrates besides digestibility, such as the types of food fibre found in foodstuffs and the levels of bioactive compounds contained therein [87]. Substitutes for the concept of GI proposed by health nutrition researchers include a new method for classifying starch digestion by modeling amylolysis of plant foods using first-order enzyme kinetic principles. This research opens new horizons and supports the relationship between levels of resistant starch, dietary fibre, phenolic, flavonoids, and the value of Food GI.

2. We have:

delete hyphen between dia-betes

delete hyphen between con-sumes

delete hyphen between sys-tem

delete hyphen between responsi-ble

etc.

3. We have deleted it

Line 400-411

Line 40

Line 46

Line 47

Line 52

etc.

-

Round 2

Reviewer 3 Report

Minor point: in the new (and welcome) discussion on GI, "some scientist" should be "some scientists"